# Gene Therapy with Voretigene Neparvovec Improves Vision and Partially Restores Electrophysiological Function in Pre-School Children with Leber Congenital Amaurosis

**DOI:** 10.3390/biomedicines11010103

**Published:** 2022-12-30

**Authors:** Maximilian J. Gerhardt, Claudia S. Priglinger, Günther Rudolph, Karsten Hufendiek, Carsten Framme, Herbert Jägle, Daniel J. Salchow, Andreas Anschütz, Stylianos Michalakis, Siegfried G. Priglinger

**Affiliations:** 1Department of Ophthalmology, Ludwig-Maximilians-University Munich, Mathildenstraße 8, 80336 München, Germany; 2University Eye Hospital Hannover Medical School, Carl-Neuberg-Straße 1, 30625 Hannover, Germany; 3Department of Ophthalmology, University Hospital Regensburg, Franz-Josef-Strauß-Allee 11, 93053 Regensburg, Germany; 4Department of Ophthalmology, Charité-Universitätsmedizin Berlin, Corporate Member of Freie Universität Berlin, Humboldt-Universität zu Berlin, and Berlin Institute for Health, Campus Virchow Klinikum, Augustenburger Platz 1, 13353 Berlin, Germany

**Keywords:** gene therapy, voretigene neparvovec, RPE65, leber congenital amaurosis, early-onset retinal dystrophy, electrophysiology, vision improvement, gene therapy in children

## Abstract

Leber congenital amaurosis caused by mutations in the RPE65 gene belongs to the most severe early-onset hereditary childhood retinopathies naturally progressing to legal blindness. The novel gene therapy voretigene neparvovec is the first approved causative treatment option for this devastating eye disease and is specifically designed to treat RPE65-mediated retinal dystrophies. Herein, we present a follow-up of the youngest treated patients in Germany so far, including four pre-school children who received treatment with voretigene neparvovec at a single treatment center between January 2020 and May 2022. All patients underwent pars plana vitrectomy with circumferential peeling of the internal limiting membrane at the injection site and subretinal injection of voretigene neparvovec. Pre- and postoperative diagnostics included imaging (spectral domain optical coherence tomography, fundus autofluorescence, fundus wide-angle imaging), electrophysiologic examination (ERG), retinal light sensitivity measurements (FST) and visual acuity testing. Behavioral changes were assessed using a questionnaire and by observing the children’s vision-guided behavior in different levels of illumination. All children showed marked increase in vision-guided behavior shortly after therapy, as well as marked increase in visual acuity in the postoperative course up to full visual acuity in one child. Two eyes showed partial electrophysiological recovery of an ERG that was undetectable before treatment—a finding that has not been described in humans before.

## 1. Introduction

In 2018, voretigene neparvovec-rzyl (VN) was launched for gene therapy of RPE65-linked inherited retinal dystrophies (IRD) in the European Union. RPE65-linked IRD account for about 3–16% of Leber congenital amaurosis (LCA2) and approximately 0.6–6% of retinitis pigmentosa (RP20) [1]. Until the advent of RPE65 gene therapy, the disease naturally progressed to legal blindness [2]. VN is the first therapeutic compound that achieved the approval of both the FDA and the EMA for an in vivo gene therapy of an eye disease and is indicated for patients with confirmed biallelic RPE65 mutation-associated retinal disease with sufficient remaining viable retinal cells [3,4,5]. RPE65-linked IRD typically manifests as Leber congenital amaurosis (LCA) or early-onset severe retinal dystrophy (EO[S]RD), characterized by severe visual impairment from birth or early infancy with light staring and profound nyctalopia accompanied by nystagmus and poor pupillary light responses [6]. A non-detectable or highly attenuated electroretinogram (ERG), as well as severely reduced or absent autofluorescence, are key diagnostic features [7,8].

The RPE65 gene is expressed in the retinal pigment epithelium (RPE) and encodes a 65-kDa isomerase that participates in the visual cycle. Since rod photoceptors are critically dependent on RPE65 as a source of 11-cis retinol, RPE65 protein deficiency leads to severe visual impairment, especially under low-light conditions, and induces early degeneration of rod photoreceptor cells [9]. To date, more than 250 RPE65 gene mutations associated with RPE65-related disease phenotypes are known, and the clinical course of the disease is believed to be critically dependent on their severity and residual enzyme activity (Human Gene Mutation Database HGMD professional 2022.2, last accessed August 7, 2022). While most patients develop symptoms within the first years of life, few patients have normal or near-normal visual acuity at young ages [1,10]. In early stages, fundus examination may be normal. Later on, signs of a pigmentary retinopathy appear with progressive degeneration, while residual visual function inevitably deteriorates towards blindness during adolescence or early adulthood. By the fourth to fifth decade, nearly all patients meet WHO criteria for blindness [1,2,10].

Voretigene neparvovec is an adeno-associated virus vector (AAV)-based subretinal gene therapy that contains 1.5 × 10^11^ vector genomes (vg) in 300 ul. Animal models of RPE65 deficiency had shown that augmentation of RPE65 can improve retinal and visual function, as assessed by means of electroretinography (ERG) and vision-guided behavior ([11], murine, in utero; [12] dog; [13], dog). Clinical trials then demonstrated a dose-dependent increase in retinal sensitivity as measured by FST and mobility test performance, although not in all participants [14]. However, so far, its magnitude could never be detected by means of ERG once the ERG before treatment had been unrecordable, not even in the youngest study participants [14,15,16]. To date, neither the magnitude nor durability of benefit reported in humans matched those observed in animal models. In this study, we describe four preschool patients with RPE65-LCA treated with VN, who showed a marked increase in BCVA up to full visual acuity in one child, and evidence for electrophysiological recovery of an ERG that was undetectable before treatment in two eyes. Clinical characteristics of the patients, genetics, surgical factors, and outcomes are discussed. 

## 2. Materials and Methods

### 2.1. Patient Selection and Overview on Performed Examinations

A retrospective chart review of four patients (*n* = 8 eyes) with molecularly confirmed *RPE65* mutation-associated Leber congenital amaurosis who were treated with VN between January 2020 and May 2022 at the Department of Ophthalmology at Ludwig-Maximilians-University (LMU) Munich was performed. The parents of the patients were informed in detail about the study and formally agreed to participate by providing their written informed consent. The study was approved by the Institutional Review Board and complied with the tenets of the Declaration of Helsinki.

All patients underwent complete ophthalmological examinations before and after treatment, including best corrected visual acuity (BCVA), intraocular pressure (IOP) measurements, slit lamp examination, funduscopy, orthoptic assessment, spectral domain optical coherence tomography (SD-OCT) and 488 nm fundus autofluorescence (FAF) using the Heidelberg Spectralis^®^ OCT (Heidelberg Engineering, Heidelberg, Germany). Goldmann visual field (VF) testing and Ganzfeld electroretinography (Toennies Multiliner Vision^®^, Weißenfels, Germany; RETI-port, Roland Consult^®^, Brandenburg an der Havel, Germany) were performed before treatment and after 6 and 12 months of treatment. To document fundus appearance, ultrawide-field fundus photographs were taken at baseline and at all post-treatment visits (California^®^, Optos plc, Dunfermland, Scotland, UK). Postoperative clinical evaluations were done in all patients at days 1, 2 and 3 after the injection; 1, 3 and 6 months after surgery (*n* = 8 eyes); and again after 12 months (*n* = 5 eyes), 24 months (*n* = 4 eyes), 27 months (*n* = 3 eyes), 30 months (*n* = 2 eyes) and 32 months (*n* = 1 eye). 

### 2.2. Gene Therapy Procedure and Perioperative Treatment 

VN was administered under general anesthesia via 23 G three-port vitrectomy by the same surgeon (S.G.P.) according to the protocol described by Russel et al. [17] with minor modifications: after induction of posterior vitreous detachment and removal of residual vitreous (every vitrectomy involved 4 mg triamcinolone acetonide (TriamHEXAL, Hexal AG, Holzkirchen, Germany) to visualize the vitreous), the internal limiting membrane (ILM) was stained using trypan blue and brilliant blue G (membraneblue® dual syringe, D.O.R.C., Zuidland, Netherlands) and peeled off at the site of retinotomy in order to reduce resistance during retinotomy, and intraocular pressure was lowered to 10 mmHg to facilitate the detachment of the retina during subretinal injection. Intraoperative OCT (Zeiss RESCAN 700, OPMI Lumera, Carl Zeiss Meditec AG, Jena, Germany) was performed in 2 of 4 patients to confirm correct subretinal delivery and to rule out any damage of the retina during the procedure. Subretinal administration of VN was performed manually with the aid of an assistant surgeon (M.J.G.) using a 41 G subretinal injection cannula (D.O.R.C., Zuidland, Netherlands) without induction of a BSS pre-bleb. It was aimed at a detachment of the retina of the whole posterior pole including the foveal area. All four patients were treated with a subretinal injection of 1.5 × 10^11^ vector genomes (vg) voretigene neparvovec-rzyl (Luxturna™) in 300 µL [3]. The patients were asked to remain in supine position for 24 h after surgery. This may minimize mechanical stress and therefore prevent potential damage to the retina, as well as potential unwanted efflux of vector solution through the retinotomy into the vitreous cavity, and to maximize contact between vector solution and RPE. The worse-seeing eye was treated first, which was determined either by visual acuity (P1, P4) or, if the initial visual acuity was equal, by subjective preference and/or parental observation, respectively (P2, P3). The second eye was injected 35–216 days after the first injection (mean 90; ±73 [SD] days). Patients received 1 mg/kg/day prednisolone (per os) beginning 3 days prior to treatment and for a total of seven days, followed by another five days with 0.5 mg/kg prednisolone which was then tapered to 0.5 mg/kg prednisolone every second day over five days. The perioperative systemic immunosuppressive regimen used in the children corresponds to that recommended in the manufacturer’s guidelines for usage of VN. Postoperative topical treatment consisted of prednisolone–acetate eye drops (Predni-POS^®^ 1%, URSAPHARM Arzneimittel GmbH, Saarbrücken, Germany) which were administered six times a day for 1 week, then tapered over a total of 6 weeks by decreasing by one drop every week. Moxifloxacin eye drops (VIGAMOX^®^ 5 mg/mL, Novartis Pharma, Nürnberg, Germany) were given four times a day for 5 days and tropicamid eye drops twice a day for one week (Mydriaticum Stulln^®^ UD, Stulln Pharma GmbH, Stulln, Germany). 

### 2.3. Efficacy Parameters/Outcome Measures

Best corrected visual acuity (BCVA) was measured by LEA charts in two patients at baseline (P1 and P2). In Patient 1 (P1, 4-year-old male) shortly after therapy, acuity assessment using Landolt C rings (single optotypes) was possible and was performed during follow-up visits. Patient 2 (3-year-old male) was examined with LEA charts during the whole period due to his young age. BCVA assessments in patient 3 (P3, 5-year-old male) were possible using a standard adult visual acuity chart with numbers. Patient 4 (P4, 6-year-old male) could be followed using Landolt C rings (single optotypes) throughout the whole observation period. BCVA was recorded in decimal and converted into LogMAR. Goldmann kinetic perimetry data were collected using stimulus test size III4e. The targets were presented at every 15° at an approximate angular velocity of 4°/s. Navigation was measured at five different luminance levels (2 lux, 4 lux, 40 lux, 100 lux and 400 lux, respectively). Within the parkour, patients were encouraged to find an object with low contrast (measurements were performed using a Luxmeter certified for clinical trials, LUX-1335, ISO-TECH, RS Components, Corby, UK). 

### 2.4. Visual Function Questionnaire (NEI-VFQ25)

For subjective assessment of treatment efficacy, patients and their parents, respectively, were asked to complete the National Eye Institute Visual Function Questionnaire-25 (NEI-VFQ25) prior to and 6 months after treatment [18]. Using this questionnaire, we aimed at evaluating changes in vision and vision-related quality of life after gene therapy to reach a more accurate picture of the treatment efficacy of VN. The questionnaire consists of 3 parts spanning 25 items plus an appendix of optional additional questions and is a psychometric tool for evaluating different aspects of vision, vision-guided behavior, and quality of life. Calculation of the overall composite score was performed according to published guidelines, as suggested by the inventors of the questionnaire [18]. Detailed description and scoring instructions of the VFQ25 can be found online (https://www.rand.org/health-care/surveys_tools/vfq.html accessed on 4 November 2022). Briefly, original numeric values from the survey were re-coded so that each item was converted to a 0 to 100 scale, with 0 set as the lowest and 100 set as the highest possible score. In this format, scores represent the achieved percentage of the total possible score so that a higher score represents better functioning and reflects better visual ability. By combining individual questions, the questionnaire creates the following eleven vision-targeted subscales: global vision rating, difficulty with near vision activities, difficulty with distance vision activities, limitations in social functioning due to vision, role limitations due to vision, dependency on others due to vision, mental health symptoms due to vision, driving difficulties, limitations with peripheral and color vision, and ocular pain. In addition, a single general health rating question has been added. Averaging mean scores of the subscale scores determines the overall composite score. In our analysis, those items that did not apply to our population or could not be answered precisely due to age were excluded. Therefore, questions on driving (items 15 and 16), mental health (items 3, 21, 22, 25, A13), and ocular pain (items 4 and 19) were excluded. Furthermore, in the calculation of the specific subscore for near vision, items A4 (“because of your eyesight, how much difficulty do you have figuring out whether bills you receive are accurate?”) and A5 (“because of your eyesight, how much difficulty do you have doing things like shaving, styling your hair, or putting on makeup?”) were excluded since they did not properly apply to pre-school children. For a similar reason, question 25 of part 3 (“I worry about doing things that will embarrass myself or others, because of my eyesight” was not considered. 

For visualization of part 2 and part 3 of the questionnaire, the original responses to each question were assigned ordinal rank scores ranging from 0 to 4 according to established evaluation models [19,20] to allow for grading and comparability of the scores throughout the questionnaire. Using this score, 4 represents the most positive response one would expect from a healthy person with normal eyesight, and 0 represents the most negative response, which would reflect the response of a blind person. 

### 2.5. Electroretinography (ERG)

Full-field ERGs (Toennies Multiliner Vision^®^, Weißenfels, Germany; RETI-port, Roland Consult^®^, Brandenburg an der Havel, Germany) were recorded before and after treatment as described previously [21] and according to International Society for Clinical Electrophysiology of Vision (ISCEV) standards [22] with the following modifications. Because of the difficulty in performing ERG measurements in pediatric patients and the discomfort to the patient, only light-adapted 30 Hz Flicker ERG was assessed with an intensity of 2.5 cds/m^2^. Preoperatively, ERG was able to be performed in 3 of 4 patients (P1, P3, P4). Postoperatively, P4 refused the repeat of the exam. In P2, Flicker-ERG could be obtained with reduced quality after the therapy. 

### 2.6. Full-Field Light Sensitivity Testing (FST)

Retinal sensitivity testing to achromatic (white) light flashes presented in the dark was measured in 30 min dark-adapted eyes using the Diagnosys Espion system with the ColorDome™ LED full-field stimulator (Diagnosys LLC, Lowell, MA, USA). Testing involved a preprogrammed setting in the Espion system, wherein a 0-decibel (dB) reference point is chosen, which for our study was 0.01 cd.s/m^2^ (25 cd/m^2^ presented for 4 ms). After defining a starting value, the proprietary software used a forced selection strategy to test within a 10 dB range around the start value. The test–retest variability of the FST was reported as 0.3 log, and a meaningful change was considered to be 10 dB or 1 log [23]. A button box was used to indicate if a brief stimulus was perceived. 

### 2.7. Optical Coherence Tomography (SD-OCT) and 488 nm Fundus Autofluorescence (FAF)

Retinal cross sections were obtained with spectral-domain optical coherence (SD-OCT) tomography using the Heidelberg Spectralis^®^ OCT (Heidelberg Engineering, Heidelberg, Germany); a 488 nm fundus autofluorescence was obtained using the same device.

### 2.8. Fundus Wide-Angle Imaging 

Fundus images were taken using the Optos California^®^ device (California^®^, Optos plc, Dunfermland, Scotland, UK) pre- and postoperatively at every follow-up visit. Images were exported in jpeg format and arranged in Adobe InDesign^®^ without further image processing.

### 2.9. Statistical Analysis 

Statistical analysis was performed using GraphPad Prism9 (Version 9.4.1) employing one-way analysis of variance (ANOVA) followed by Dunnett’s multiple comparison test. Data gained by the NEI-VFQ25 was further analyzed using the statistical software R (version 4.2.2). Exported figures were arranged using Adobe InDesign^®^ without further processing. 

## 3. Results

### 3.1. Patient Demographics and Genetic Background 

All patients included in this report had confirmed biallelic RPE65 mutations and presented with the clinical picture of the most severe form of RPE65 IRD, Leber congenital amaurosis. The four patients were all young males and were 3, 4, 5 and 6 years old at the time of treatment. RPE65 mutations were compound heterozygous in three and homozygous in one patient (Table 1).

Patient 1 (P1), a 4-year-old male of southeast European heritage harbored two heterozygous mutations in RPE65. One of the two variants, namely a 4-base pair (bp) duplication (c.1207.1210dupCTGG) in Exon 11 has already been described in the literature once in the context of autosomal-recessive retinitis pigmentosa [24]. It leads to a shift of the open reading frame and hence causes the formation of a premature stop codon, p.(Glu404Alafs*4). Furthermore, a c.1338+1G>A mutation in Intron 12 of the RPE65 gene was detected in a heterozygous state. This mutation affects the highly conserved splice donor site of Exon 12, which, according to bioinformatic analysis, should lead to skipping Exon 12. To the best of our knowledge, the c.1338+1G>A mutation has not yet been described in the literature. However, there is currently one entry in the ClinVar database that classifies the mutation as pathogenic. Both sequence changes do not occur in the control population worldwide.

Patient 2 (P2), a 3-year-old male of middle European (German) ancestry harbored two pathogenic heterozygous mutations in RPE65 and one variant in the CEP290 gene with unclear clinical significance regarding its disease phenotype. The c.74C>T missense mutation in Exon 2 of RPE65 should lead to replacement of a highly conserved amino acid (p.(Pro25Leu)) and is very rare in the control population (4 of >250,000 alleles, MAF 1.6 × 10^−5^). The variant is described in the literature as a hypomorphic mutation with a mild phenotype in a homozygous state [25]. Functional studies indicate that the mutant protein is misfolded and rapidly degraded by the proteasome [26]. The second mutation found in RPE65, c.1543C>T in Exon 14, is another missense mutation leading to the replacement of a highly conserved amino acid (p.Arg515Trp) and is very rare in the control population (4 of >280 000 alleles, MAF 1,4 × 10^−5^). It is described several times in the literature with autosomal-recessive RP/LCA (e.g., [27,28]). Functional studies demonstrate strongly reduced enzyme activity of the mutant protein. Apart from the two RPE65 missense mutations, the c.5668G>T variant in Exon 41 of the Centrosomal Protein 290 (CEP290) gene was detected in P2 in a heterozygous state. In silico, this variant leads to the formation of a premature stop codon with the resulting mRNA being degraded via nonsense-mediated decay. This variant is described several times in connection with autosomal-recessive retinopathies such as RP, Joubert syndrome and other ciliopathies (e.g., [29]). Since no second variant was detected, and the two variants in RPE65 explain the disease phenotype of RPE65 mutation associated IRD, it is unlikely that the variant in CEP290 contributes to the IRD in P2. However, no definite statement on the exact clinical significance can be made. 

Patient 3 (P3), a 5-year-old male of southeast European heritage carried the familial variant c.1207_1210dupCTGG in RPE65 in a homozygous state. As described above, the duplication of four nucleotides creates a shift in the reading frame, resulting in a premature stop-codon (p.Glu404Alafs*4), which will likely result in nonsense-mediated decay of the mRNA transcript. The alteration has previously been described in literature compound heterozygous with another variant in RPE65 in association with retinitis pigmentosa in several affected family members ([24], reported as Glu404 (4-bp ins) (GAG to GCTGGAG)).

Patient 4 (P4), a 6-year-old male of middle European (German) heritage showed compound heterozygosity, which was confirmed with segregation analysis of his parents. The first detected variant, a heterozygous c.11+5G>A mutation in Intron 1 of the RPE65 gene is very rare in the control population worldwide (22 of> 280,000 alleles, MAF 10^−5^) and affects the highly conserved splice donor site of Exon 1, so that at least part of the RPE65 transcripts should not be spliced correctly. To the best of our knowledge, experimental data for such an effect are lacking. In the ClinVar database, there are currently more than 10 entries listing this variant as (probably) pathogenic. In the literature, there are several reports of the c.11+5G>A mutation in connection with RP/LCA [30]). The second heterozygous variant detected in patient 4, c.726-2A>T in Intron 7, does not occur in the control population worldwide and affects the highly conserved splice acceptor site of Exon 8, which, according to bioinformatic analysis, should lead to skipping of Exon 8. To the best of our knowledge, experimental data for such an effect are not yet available. In the ClinVar database, two entries are currently listed, which assess the c. 726-2A> T mutation as (probably) pathogenic, and in the literature, it has already been described in connection with the development of LCA (e.g., [31]. Table 1 summarizes the patient demographics and genetic features.

### 3.2. Baseline Characteristics and Clinical Appearance 

All patients except for P4 showed a lack of eye contact from birth and had infantile nystagmus, which was first perceived by their parents at different times as detailed in Table 2. All patients showed different degrees of hyperopia with mild astigmatism. Baseline/pretreatment BCVA ranged from 1.3–0.7 logMAR (mean 1.01; SD ± 0.27). Distinctive of the disease, a lack of 488 nm autofluorescence was observed in all patients. Navigation at luminance levels below 40 lux was not possible, and none of the patients was able to reliably disclose the III/4 isoptere in Goldmann visual field testing before treatment. Mean time of follow-up was 18.5 months (range 6–32 months; SD ± 10.4). 

### 3.3. Local Distribution of Voretigene Neparvovec

Subretinal administration of voretigene neparvovec involving the posterior pole including the foveal region was successfully performed in all eyes. This required a minimum of one and a maximum of four retinotomies through which the subretinal injection was performed. The primary injection site was selected at the superior vascular arcade to comprise the macular and perifoveal regions as recommended in the manufacturer’s instructions. In case of centrifugal spread of the subretinal fluid, more than one retinotomy was necessary to treat sufficient retina at the posterior pole. Figure 1 shows the intraoperative view after subretinal injection (images are inverted and laterally reversed just like the view through the surgical microscope). Following fluid–air exchange, the area of treated retina was further enlarged in two patients (both eyes of P3 and P4) by enlarging the bleb laterally. This was achieved through gentle suction at the margins of the bleb using a backflush flute needle.

### 3.4. Immediate Postoperative Course and Adverse Events

In the early postoperative phase (up until 4 weeks), ophthalmological follow-up of six of eight eyes revealed no deleterious effect, and the postoperative course was similar in all seven eyes with no evidence of severe intraocular inflammation. Mild, transitory intraocular inflammation, as well as conjunctival hyperemia, was evident after surgery, as typically seen after vitrectomy, and resolved within 4 weeks under topical treatment with corticosteroids. Subretinal fluid was absorbed within 24–48 h. 

Patient 3 developed rhegmatogenous retinal detachment with macular detachment at day 7 after surgery in his right eye. After retinal re-attachment surgery involving silicon oil and removal of the silicon–oil tamponade four months after implantation, the retina remained soundly attached until the last follow-up. Despite this complication, visual acuity increased to 0.9 logMAR (0.16 decimal) in the postoperative course, as seen at month 6 after subretinal gene therapy. However, cataract formation was observed, which was presumably responsible for the decrease in visual acuity to baseline values. Patient 4 developed increased intraocular inflammation 1 week after surgery of the second (OD) eye, which clinically appeared as mild non-infectious vitritis. The inflammation fully resolved within 8 weeks after intensifying topical treatment with corticosteroid eye drops. In the same eye, as observed at a routine check-up 3 months after therapy, a circumscribed peripheral retinal detachment was apparent as an incidental finding, which the patient had not noticed. Due to the rhegmatogenous origin, vitrectomy involving retinal cryopexy and sulfur hexafluoride (SF_6_) gas endotamponade had to be performed. In P1, 4 weeks after surgery on the right eye, a slight roundish irregularity at the inferior vascular arch was first clinically observed, which later turned out to be a circumscribed atrophy of the photoreceptor layer at the injection site (see Figure 2). The lesion has remained unchanged since it was first noticed, with no tendency to increase over a follow-up time of 32 months. In light of recent disturbing reports of progressive chorioretinal atrophy development following gene therapy with voretigene neparvovec [32,33], we took a close look at our own data and could not observe any atrophy development. 

### 3.5. Efficacy of Voretigene Neparvovec

#### 3.5.1. Best Corrected Visual Acuity (BCVA)

The results of best corrected visual acuity are given in logMAR units wherein smaller values indicate better acuity. A 0.1 increment in logMAR corresponds to five letters or one line on the ETDRS chart. In clinical studies, a well-accepted level of clinical significance is the improvement of three or more lines (equals to 15 letters on the ETDRS chart), which corresponds to a 0.3 logMAR change. At baseline, BCVA ranged from 1.3–0.7 logMAR (mean 1.01; SD ± 0.27). Four weeks after surgery, mean visual acuity was 0.85 logMAR (SD ± 0.34), and at the six month follow-up, we observed a statistically (adjusted *p*-value 0.01) and clinically significant mean visual acuity improvement of 0.3125 logMAR units (mean BCVA 0.7; SD ± 0.3). Notably, six eyes of three patients reached an improvement of at least 0.3 logMAR units, with the highest gain in the right eye of the three-year-old child (P2 OD) to an extent of 0.5 logMAR units. This corresponds to an improvement of five lines (25 letters) on the ETDRS chart and is particularly remarkable since the three-year-old boy presented with the worst visual acuity at baseline, considering that assessment of BCVA was only possible binocularly and in the near before treatment. To ensure comparability of the data set, a statistical analysis was only carried out up to the 6-month follow-up (*n* = 8 eyes, see Table 3). However, looking at later follow-ups of individual eyes, we were delighted to confirm marked further improvement of visual acuity in two patients. Thirty-two months after gene therapy of the first (left) eye, P1 presented with a visual acuity of 0.6 logMAR in the left and 0.5 logMAR in the right eye. This corresponds to an improvement of 0.5 (OD) and 0.7 logMAR units (OS), respectively. Comparable observation was made for P4. Despite another vitrectomy due to retinal detachment of the right eye, BCVA improvement of 0.5 logMAR units was observed at month 6. Eight months after surgery, the contralateral eye (OS) showed full visual acuity (0.0 logMAR; 1.0 decimal) representing a −0.7 logMAR (35 letters on ETDRS chart) improvement. For an illustration of the individual courses of visual acuity of all treated eyes up to the last follow-up, see Figure 3. Table 3 shows mean changes of visual acuity from baseline at month 1 and 6 in both eyes. All participants except P4 had infantile pendular nystagmus with medium to high frequency before treatment. In all of them, nystagmus dampened or had become yerk-type nystagmus with reduced amplitude at 6 months after gene therapy. For a complete overview of the visual acuity values collected over the whole observation period and comparison to baseline values, see Appendix A.

#### 3.5.2. Electroretinography (ERG)

Preoperatively ERGs (P1, P3 and P4) and 30 Hz Flicker-ERG response in particular were unrecordable in all examined patients except for background noise, which was visible upon magnification. ERG of P2 could not be recorded prior to therapy because of non-compliance due to young age and discomfort caused by lid electrodes. P4 refused to repeat ERG recording after treatment. Six months after treatment, the right eye of P2 showed reproducible 30 Hz Flicker responses. Furthermore, 30 Hz flicker ERG in patient 1 and patient 3 revealed reproducible 30 Hz Flicker amplitudes in one of their eyes (P1 OD and P3 OS, see Figure 4), which had not been detectable before treatment. Appendix A shows Flicker ERGs of P2, which were recorded 6 months after gene therapy. The right eye showed recordable 30 Hz flicker responses, whereas the left eye did not show clear evidence of electrophysiologic cone response. Unfortunately, no follow-up recording could be performed because of the patient’s resistance and discomfort caused by lid electrodes and the parent’s disapproval of the examination. 

#### 3.5.3. Full-Field Light Sensitivity Testing (FST)

Due to the demanding experimental set-up, the young age of the patients and the associated reduced cooperation and attention span, we only succeeded in taking reliable measurements in two patients (P3, P4). P3 could not perform the test reliably before treatment, but measurements were obtained 1 week and 6 months after surgery of the left eye. The right eye was first tested approximately 8 months after initial gene therapy, which corresponded to 5 months after silicon oil removal and was measured again 6 months after the first examination. In P4, FST was performed prior to surgery and 6 months after treatment of the second eye. 

Whereas P3 was not able to perform the test before treatment, a threshold of −37.41 dB could be detected at the first examination approximately 8 months after subretinal gene therapy of the right eye. One year after therapy, the sensitivity had increased to −58.20 dB in the same eye (ΔdB = −20.79). The contralateral left eye was measured first 1 week after surgery showing a threshold of −40.73 dB, one week later the threshold was −28.04 dB with doubtful reliability due to reduced recording quality. At the last follow-up 6 months after therapy, a threshold of −32.12 log dB could be recorded. 

P4 showed a remarkable increase in full-field light sensitivity from −09.55 dB at baseline to −38.05 dB in the right eye 6 months after therapy (ΔdB = −28.50), despite a postoperative complication. The degree of improvement in the left eye was even higher with a 55.49-fold increase from −10.76 dB at baseline to −66.25 log dB eight months after therapy. 

#### 3.5.4. Observation of Vision Guided Behavior/Behavioral Changes 

Within days to weeks after therapy, all children reported a temporary increase in light sensitivity, followed by significantly better orientation in dim light. Parents observed a marked increase in visual fields. Two of the treated children (P1, P4) were no longer dependent on additional light sources in their homes that had become necessary before treatment. Furthermore, parents noticed a decrease in nystagmus, indicating better visual perception and fixation, as well as improvement in visual acuity. Six months postoperatively all children were able to navigate monocularly and binocularly at a luminance level of 4 lux as opposed to 40–400 lux before treatment. 

#### 3.5.5. Visual Function Questionnaire 

To reach a more accurate picture on the impact of gene therapy, parents were asked to respond to the National Eye Institute Visual Function Questionnaire-25 (NEI-VFQ25) and report changes in vision and vision-related behavior, as observed or mentioned by their child in daily life. Since the questionnaire is composed of a subset of both vision-related and general health-related items, further analysis was performed only using the items relevant for vision-guided behavior. Items that assessed activities requiring good eyesight (such as reading signs, using hand tools, or looking for something on a crowded shelf) showed a significant higher score indicating marked visual improvement after therapy. The same striking improvement was seen in the items that asked about dependencies on other people due to visual impairment. All patients or their parents stated that the children became more independent, self-reliant, and self-confident as a result of their improved vision. Figure 5 illustrates the impressive improvement in the children’s visual ability showing the reported responses to questions of part 2 and part 3 of the VFQ25 before and 6 months after gene therapy as mean scores of all items for P1–P4. The individual responses to every evaluated item of VFQ25 are illustrated in Appendix A. The mean scores of the evaluated vision-targeted subscales and the overall composite score are shown in Table 4. According to the standard evaluation method, original numeric values from the survey were re-coded so that each item was converted to a 0 to 100 scale. In this format, scores represent the achieved percentage of the total possible score so that a higher score represents better functioning and reflects better visual ability. The average improvement of 35.05 points in the score clearly demonstrates the marked improvement in the children’s visual ability after gene therapy.

#### 3.5.6. Goldmann Kinetic Perimetry

Preoperatively, none of the children was able to reliably disclose the III_4_ target in Goldmann visual field perimetry due to the young age and severe visual impairment. A usable visual-field examination could be performed in only one of the four children (P4) before treatment. However, at that time, P4 was only able to give vague information about the visual field’s outer limits with the III_4_ target. Postoperatively, reliable visual fields with a clearly measurable III_4_ isopter could be obtained in three patients (P1 OU, P3 OD, and P4 OU). P2 was three years old at assessment and too young to reliably perform visual-field testing. Figure 6 shows Goldmann visual fields of P4 both pre- and postoperatively with marked enlargement of the visual field after gene therapy. Appendix A shows Goldmann visual fields of P1 OU and P3 OD after gene therapy. At time of assessment, the left eye of P3 was not yet treated which explains the missing data.

## 4. Discussion

The present study reports a follow up of some of the youngest patients treated with voretigene neparvovec so far. For the first time, partial recovery in retinal electrical activity in two eyes of two patients with previously undetectable ERG is shown, and a marked improvement in BCVA after treatment with AAV2-RPE65 reaching full visual acuity in one patient is reported. 

Since RPE65 mutations in RPE affect rod function even at earliest disease stages, patients typically suffer from severe night blindness and severely reduced visual acuity and abnormal eye movements (nystagmus) from an early age on. For this reason, a multiluminance parcour (MLMT), full-field stimulus test (FST) and Goldmann perimetry were established as outcome measures for rod-mediated vision in the clinical trials preceding FDA approval of VN [17]. Thus, the improvement of BCVA, a measure of foveal cone-mediated function, was somewhat unexpected in our patient cohort. In terms of treatment efficacy, we found improvement in visual acuity with a mean change of >0.30 logMAR units six months after gene therapy, which is a commonly accepted criterion for clinical significance. This is remarkable with respect to the pivotal studies, wherein improvement in visual acuity was much lower in magnitude and not of clinical significance. The clinical phase 3 trial that led to FDA approval of VN reported an average gain of +8.1 letters in the ITT group compared with +1.6 letters in the control group after 1 year. Only after excluding two patients who dropped out prematurely in both study arms, the modified analysis of the ITT group reached a statistically significant yet not clinically relevant difference in favor of the treatment with an increase of +9.0 vs. 1.6 letters [17]. Accordingly, pooling data from four clinical studies on AAV-RPE65-gene therapy, a recent Cochrane analysis found a statistically significant improvement of visual acuity in treated eyes at 1 year post treatment by only −0.1 logMAR. In a subgroup analysis, this effect was only evident in patients with baseline acuity better than 1.3 logMAR but did not exceed −0.11 logMAR [34]. In those clinical studies that involved young children, gain in visual function was reported to be higher than in adults [35]. BCVA increase ranged from 6–14 letters in Weleber’s young age-group [36], but none reached the magnitude observed in our patients and none of them was as young as our patients at the time of treatment. Deng et al. published a follow up of 27 eyes of 14 pediatric patients (age range 4–17 years) with a mean visual acuity improvement of +7.5 up to +12.5 ETDRS letters [37], and Sengillo reported a mean visual acuity change in a pediatric population of 10.8 letters (SD ± 13.8) 10–15 months after gene therapy, mentioning that seven pediatric eyes reached an improvement of ≥3 lines after 1 year [38]. Recently, marked visual acuity improvement was published in six pediatric patients (age range 7–16 years) with a mean change of −0.2 logMAR (SD ± 0.07) [39]. 

Nonetheless, nystagmus may be a confounding factor for BCVA measurement. Using a serotype 4 AAV-RPE65 vector, Le Meur et al. also observed marked improvement in a subset of patients with comparable clinical characteristics, whereas patients with no nystagmus or worse visual acuity at baseline showed no alteration [40]. In their study patients presenting with nystagmus and visual acuity between 7–31 ETDRS letters (comparing to logMAR 1.6–1.1) before treatment had an average gain of 7.6 letters. They hypothesized that the better outcomes in this subgroup may be related to the clinical modification of nystagmus [40]. In the present case series, three of four participants had infantile pendular nystagmus with medium to high frequency before treatment, and in all of them, nystagmus dampened or had a markedly reduced amplitude at 6 months after gene therapy. A correlation between a reduction in the amplitude of nystagmus and increased visual acuity has been reported previously [41,42], and this was indeed what we observed in our patients. This association already has been proposed to improve visual acuity in patients with infantile nystagmus syndrome [43] and is most notably due to an increase in foveation time. Thus, high-frequency pendular nystagmus may confer an underestimation of preoperative BCVA, which therefore may not be a fully reliable marker for treatment efficacy. Conversely, the postoperative dampening of the nystagmus in three out of our four preschool children may provide indirect evidence for treatment efficacy, suggesting an increase in foveation time due to an improved BCVA. Intriguingly, P4 showed marked increase in visual acuity and behavioral vision without any history of nystagmus, which is most likely due to the direct effect of gene supplementation. 

However, quantification and exact assessment of nystagmus characteristics may serve as an additional efficacy parameter in the future. Another challenging but fascinating question is to what extent neuronal plasticity contributes to the improvement of visual acuity after gene therapy—in particular in pre-school children. It is known that both the human and canine cortex in individuals affected by RPE65 deficiency is responsive to treatment and may be used as proof of therapy response after gene therapy [44]. 

Apart from changes in vision-guided behavior and increased visual acuity, the most remarkable finding from this case series was the modification in retinal electrical activity in patients treated with voretigene neparvovec. In contrast to the data reported in preclinical studies in RPE65-/- Briard dogs, full-field ERG recordings that had been undetectable before treatment remained diminished in all clinical gene therapy trials on RPE65-associated inherited retinal dystrophy despite subjective and objective improvements in light sensitivity [44]. In our case series, full-field ERG was not recordable in all children prior to VN and yielded recordable retinal activity in three of our treated children. In Maguire’s cohort from 2009, the youngest patient was 8 years old at time of treatment. Four children aged 8–11 years showed reproducible improvement in their visual field and had substantial improvement in their ambulation in dim light, but as in older individuals, scotopic and photopic electroretinographic responses were flat before and after injection [35]. Only multifocal electroretinography in two of the children gave some evidence of photopic responses in one part of the injected retina [35]. Weleber monitored small but recordable ERG responses in 4 of 12 patients before treatment with rAAV-CB-hRPE65, out of which 2 patients showed an increase in the 30 Hz flicker response in the treated eye. However, in contrast to our two young patients, none out of those with nonrecordable ERG before treatment demonstrated convincing increase in the ERG responses in either eye after treatment [36].

An age dependence of the treatment effect has been proposed but never been convincingly confirmed. The patients treated in the present case series are among the youngest children who have ever been treated with VN worldwide. The initial study results of objective and subjective tests supported the notion that the greatest improvement in visual function with subretinal gene therapy will occur in young individuals [35]. They had the greatest overall improvement in vision in the measures MLMT, FST and VF. Gain in BCVA, however, was not clearly age-related [14,16,35,36] but rather correlated to the presence of viable photoreceptors, although a clear structure–function relationship could not be confirmed in all studies [14].

Although a clear correlation between genotypes and their response to the intervention has not yet been shown, individuals with missense mutations known to mediate isomerase activity may be more predestined for functional improvement through gene therapy. Although the extent of disease in human RPE65-IRD at different ages is not predictable, a higher number of viable cells is more frequently found at a younger age in any disease course. Nevertheless, at the cellular level in this respect, the influence of concomitant dysregulation of several cellular functions in advanced disease stages must also be taken into consideration. Undoubtedly, the underlying pathogenic mutations are the primary cause of consecutive retinal degeneration in inherited retinal dystrophies and represent the primary therapeutic target. However, the process of degeneration that culminates in cellular dysfunction and lateron cell death is the result of a cascade of dysregulated biochemical pathways that may need to be addressed as supplemental therapeutic targets to enhance the therapeutic effect of gene therapy and halt disease progression. Numerous therapeutic targets have been identified so far [45]. For example, in animal models, progressive oxidative damage to cones and the inner retina occurs after rod photoreceptors die [46]. Equally, oxidative stress of the RPE was shown to be a major contributing factor to cell death in retinitis pigmentosa and LCA [47]. As shown in recent NGS, transcriptome and epitransciptome analyses, oxidative stress elicited by the phototoxic effect of the oxidant agent N-retinylidene-N-retinyl ethanolamine (A2E) dysregulates several biochemical pathways, including response to oxidative stress, carbohydrate and lipid metabolism only shortly after induction of oxidative stress and alters posttranscriptonal editing of genes involved in cell-to-cell adhesion and signal-transduction, all of which ultimately can lead to apoptosis [48,49]. In line with a presumed role of mutation-induced oxidative stress in retinal dystrophies, antioxiants reduced oxidative damage in models of RP [50]. In a recent phase 1 clinical trial, the potent antioxidant oral N-acetylcysteine was found to be safe and well-tolerated. The phase 2 clinical trial to assess its protective effect on visual functions of patients suffering from moderately advanced RP is now underway and still recruiting (NCT04864496). Although fully speculative at this time and not adressed in this study, gene therapy for advanced stages of IRDs may require supportive treatments and results from NGS studies may help to unravel novel therapeutic targets. Due to their young age, the patients described here possibly were at such an early stage of retinal degeneration that the detrimental effects of the mutation-induced biochemical dysregulation still was at a very beginning, so that the correction of the gene defect alone was sufficient to elicit even a partial electrophysiological recovery.

## 5. Conclusions

In summary, we present evidence that gene therapy with voretigene neparvovec may restore vision to a level of previously undetectable electrophysiological activity with treatment at a young age when more viable retinal cells are still preserved. Moreover, we report visual improvement, including full visual recovery, following gene therapy in severe congenital RPE65-mediated retinal dystrophy. To the best of our knowledge, this magnitude of improvement has never been achieved before through gene therapy. Our sample size was small, but our data and growing evidence from recently published data would support the concept of early treatment to maximize treatment benefit. 

## Figures and Tables

**Figure 1 biomedicines-11-00103-f001:**
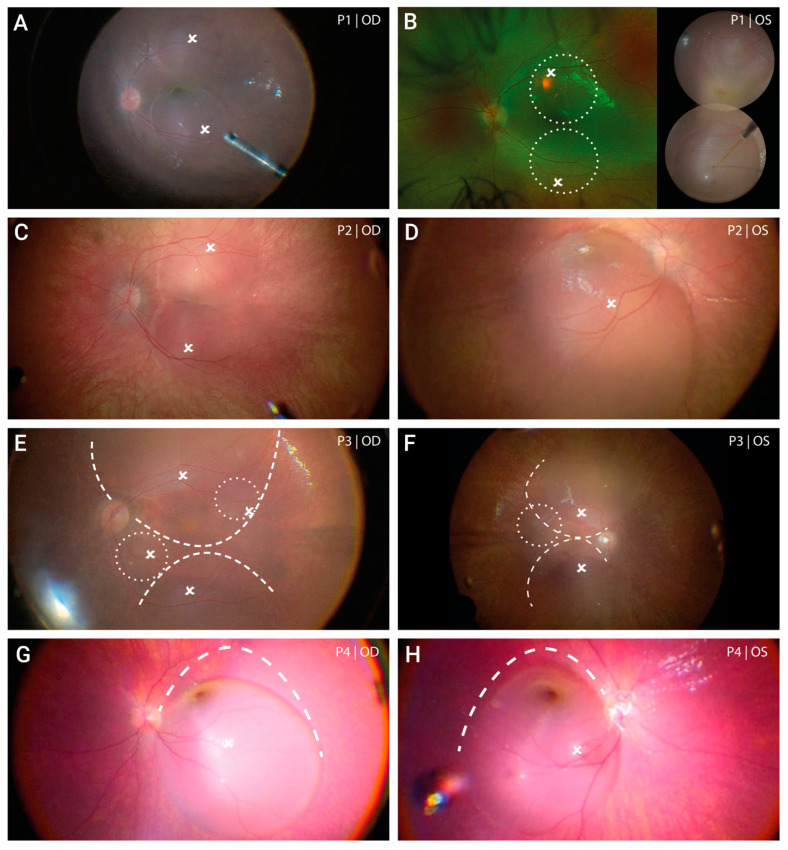
**Extent of subretinal gene therapy with voretigene neparvovec.** In both the right (**A**) and the left eye (**B**) of patient 1 two subretinal blebs were formed of which the first bleb placed at the superior vascular arch led to foveal detachment. Due to limited intraoperative records, the final extent of the subretinal bleb formation of the left eye is schematically shown by white circles placed on a pre-operative fundus image, and is supplemented by an image of the two blebs during injection. Treatment of the right eye in patient 2 (**C**) consisted of two subretinal blebs that merged at the level of the fovea (kissing blebs). In the left eye (**D**) one large bleb was formed with marked spread to the superior periphery. Subretinal injection in patient 3 was challenging due to a remarkable shift of the vector solution towards the retinal periphery. After initial injection, two smaller blebs (white circles) were added in the right (**E**) and one bleb in the left eye (**F**). In both eyes, treatment of the posterior pole could be achieved by enlarging the bleb through gentle suction at the margins of the bleb using a backflush flute needle. This procedure was performed following fluid-air-exchange. The extent of enlargement is illustrated by the dashed lines. Both the right eye (**G**) and left eye (**H**) of patient 4 was treated with one large bleb. Following injection, the bleb was enlarged to maximize the treated area using the same technique as described for patient 3. Please note: all intraoperative images (except for P1 OS) are inverted and laterally reversed (surgeon’s view). White crosses indicate the subretinal injection sites. Dashed lines indicate the final edges of the bleb after enlargement.

**Figure 2 biomedicines-11-00103-f002:**
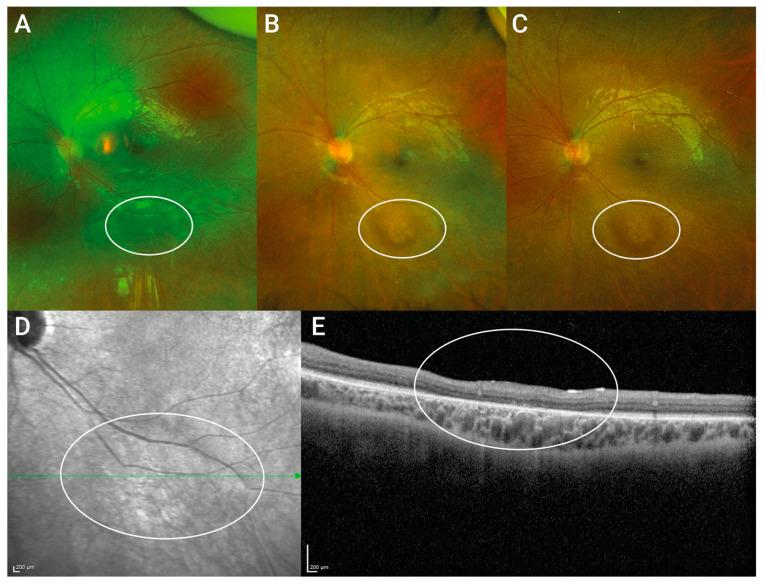
**Photoreceptor layer loss at the injection site.** Upper Panel shows postoperative wide-angle fundus images of P1 taken 4 weeks (**A**), 23 months (**B**) and 32 months (**C**) after subretinal gene therapy. The roundish irregular lesion at the injection site at the inferior vascular arcade was recognized first 4 weeks after therapy and did not extend over time. (**D**) near-infrared scanning laser ophthalmoscope fundus image 32 months after therapy shows only mild changes. (**E**) OCT through the lesion taken 32 months after gene therapy revealed circumscribed photoreceptor layer loss at the injection site but no atrophy of the choroid nor the RPE was observed.

**Figure 3 biomedicines-11-00103-f003:**
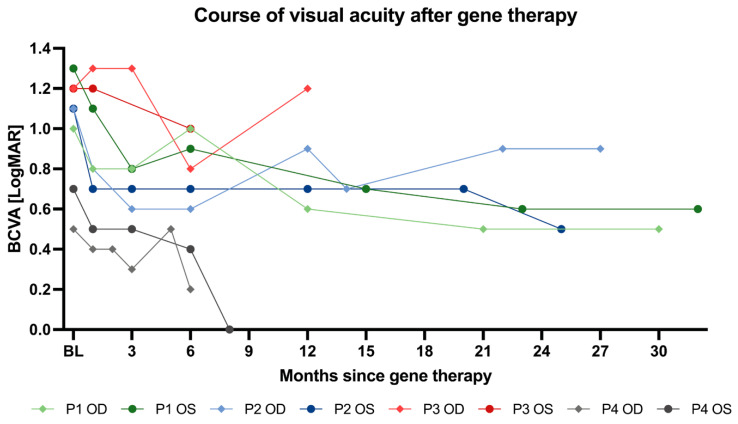
**Course of visual acuity after gene therapy with voretigene neparvovec.** Course of best corrected visual acuity (BCVA) of each treated eye is shown ranging from a minimal postoperative follow up time of 6 months (*n* = 8) up to 32 months (*n* = 1).

**Figure 4 biomedicines-11-00103-f004:**
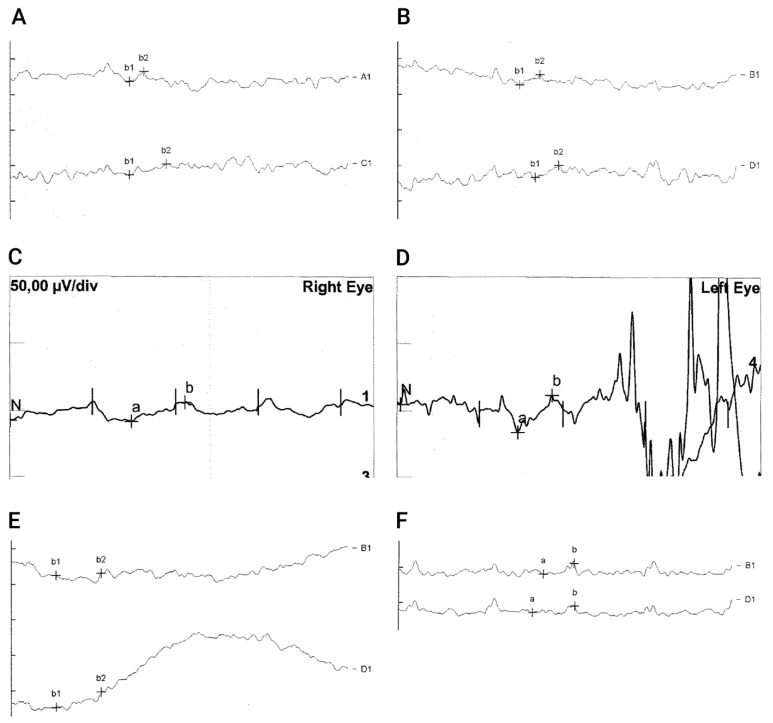
**Measurable 30 HZ Flicker-ERG after gene therapy with voretigene neparvovec.** Upper Panel (**A**,**B**) shows Flicker-ERG of a 4-year-old male (P1) taken before therapy. Both in the right (**A**) and left (**B**) eye only background noise could be recorded even at high magnification of 10X (shown is the 5X magnification). After gene therapy, 30 Hz flicker responses could be obtained in the right eye (shown is a magnification of 4X) (**C**). Flicker responses of the left eye could not be properly evaluated due to compliance-related poor recording quality (**D**). Lower panel (**E**,**F**) shows 30 Hz flicker ERG of the left eye in a 5-year-old child (P3) at 10X magnification. Before gene therapy, only background noise was recordable (**E**). After gene therapy, reproducible 30 Hz flicker responses could be recorded in the left eye (**F**).

**Figure 5 biomedicines-11-00103-f005:**
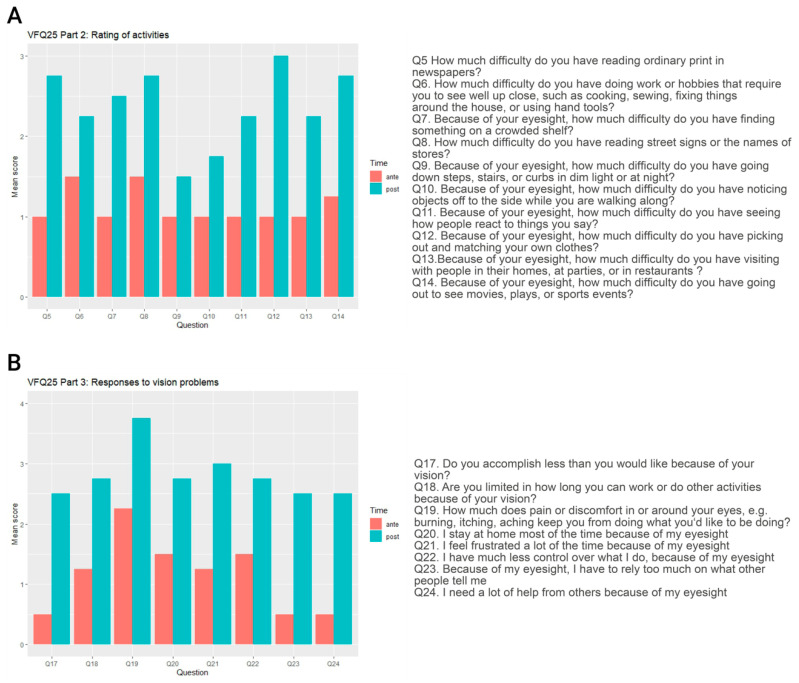
**Vision-related behavioral and psychometric changes after gene therapy.** (**A**,**B**) show the reported responses to questions of the VFQ25 on certain activities that require good eyesight (**A**) and to questions about things whose performance may be affected by vision (**B**). Responses before therapy (ante) are represented by red columns, whereas the turquoise columns represent responses given 6 months after gene therapy (post). Individual scores for each patient and item ranged from 0 (representing the worst possible answer) to 4 (most positive answer as one would expect from a normally sighted person). Illustrated are the mean scores of each item for P1–P4 (*n* = 4). All items showed a clear increase in score and thus an improvement of the visual ability was reported by all patients and their parents, respectively.

**Figure 6 biomedicines-11-00103-f006:**
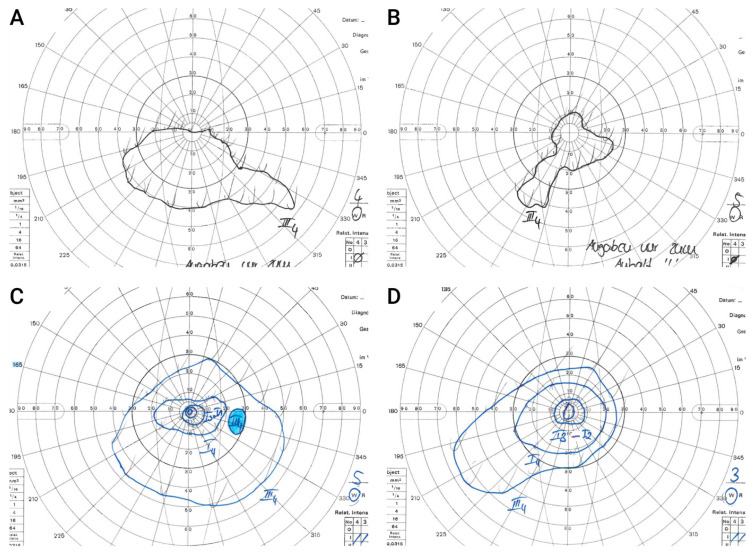
**Goldmann Visual fields in a 6-year-old child before and after gene therapy.** Upper Panel shows visual fields of a 6-year-old child (P4). Before surgery, he was able to give only vague information about the visual field outer limits in both the right (**A**) and left (**B**) eye due to severe visual impairment. After surgery, reliable field testing using targets I_1_–I_4_ and III_4_ as well as defining the blind spot was possible in the right eye 6 months after treatment (**C**). In the left eye, visual field testing using targets I_2_, I_3_ and III_4_ could be performed 8 months after therapy (**D**). The illustrated isopters correspond to the aforementioned targets and represent contour lines of the hill of vision. Both eyes showed marked enlargement of the visual field after gene therapy.

**Table 1 biomedicines-11-00103-t001:** Patient demographics and genetic features.

Patient Age at Time of Treatment, Sex	Mutations in RPE65	Type of Mutation	Functional Consequence
P1, 4 years male	Exon 11 c.1207.1210dupCTGG	duplication, frameshift	p.(Glu404Alafs*4)→stop codon, non-sense mediated decay of mRNA **
	Intron 12 c.1338+1G>A	splicing donor site	→skipping of Exon 12 **
P2, 3 years male	Exon 2 c.74C>T	missense	p.(Pro25Leu), loss of function
	Exon 14 c.1543C>T	missense	p.(Arg515Trp), loss of function
	*+ CEP 290 Exon 41 c.5668G>T **		*p.(Gly1890 *)→stop codon, non-sense mediated decay of mRNA ***
P3, 5 years male	c.1207_1210dupCTGG homozygous	duplication, frameshift	p.(Glu404Alafs*4)→stop codon, non-sense mediated decay of mRNA **
P4, 6 years male	Intron 1 c.11+5G>A	splicing donor site	→incomplete splicing of transcript **
	Intron 7 c.726-2A>T	splicing acceptor site	→skipping of Exon 8 **

* additional pathogenic variant in CEP 290 in heterozygous state; ** bioinformatic prediction, experimental results lacking.

**Table 2 biomedicines-11-00103-t002:** Baseline characteristics of the patients.

Symptom/Objective	P1 (4y)	P2 (3y)	P3 (5y)	P4 (6y)
Lack of eye contact	at birth	at birth	at birth	no
Nystagmus (Time of appearance)	few days postnatal	6 months postnatal	6 weeks postnatal	not observed
Inability of orientation in dim light *	6 months	from birth	from birth	from birth
Discomfort in the dark *				
OD	1.0 1.3 -	Fix	1.2	0.5
BCVA OS		Fix	1.2	0.7
(LogMAR) OU		1.1 **	-	-
Refractive error OD OS	+4.00/−0.25/165 +4.00/−0.50/30	+1.00/−1.50/145° +1.00/−1.50/145°	+4.50/−0.50/180° +4.50/−0.50/5°	+2.25/−1.00/170° +2.25/−1.00/170°

* observed by parents in daily routine; ** baseline near visual acuity assessment was possible only binocularly; upon occlusion of either eye fixation with the partner eye was observed.

**Table 3 biomedicines-11-00103-t003:** Changes of best corrected visual acuity from baseline up to 6 months.

Timepoint	BCVA (LogMAR)	Mean Change from Baseline (LogMAR Units)
Baseline	1.01 ± 0.27	
Month 1	0.85 ± 0.34	0.1625
Month 6	0.70 ± 0.30	0.3125

Visual acuity data reported as mean visual acuity of all 8 treated eyes ± standard deviation; change from baseline is reported in LogMAR units; change of BCVA reached statistically significant levels (adjusted *p*-value 0.01) at month 6.

**Table 4 biomedicines-11-00103-t004:** VFQ25 subscale scores and overall composite score before and after treatment.

VFQ25 Subscale	Mean Score Pre-Operative	Mean Score Post-Operative	Mean Change
General vision	13.75	49.375	+35.625
Near vision/activities	28.13	62.50	+34.375
Distance vision/activities	31.25	64.58	+33.33
Visual specific			
-social functioning	33.33	66.66	+33.33
-role difficulties	25.00	57.81	+32.81
-dependency	23.44	65.63	+42.19
Color vision	25.00	75.00	+50.00
Peripheral vision	25.00	43.75	+18.75
Overall composite score	25.61	60.66	+35.05

## Data Availability

The authors confirm that the most relevant data supporting our findings in this study are available within the manuscript and its Appendix A. Additional data related to the data published in this manuscript or supporting our findings are provided by the corresponding author (S.G.P.) upon reasonable request.

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
