# Peer review of "Gene Therapy with Voretigene Neparvovec Improves Vision and Partially Restores Electrophysiological Function in Pre-School Children with Leber Congenital Amaurosis"

_biomedicines, 2022, doi:10.3390/biomedicines11010103_

Round 1

Reviewer 1 Report

Gerhardt et al describe the results of Voretigene Neparvovec (RPE65 gene therapy) in four children afflicted with Leber congenital amaurosis (LCA). The follow-up results in these pre-school aged children showed improvement in vision and improved results across all retinal electrophysiology assessments.  This is a well-written manuscript and provides important information about the responses and outcomes of RPE65 gene therapy in children. Following are the comments on the manuscript:

Major comments:

1.       One aspect that is challenging to following the manuscript is the data for individual patients and why some of the measurements are missing. Most of it is stated in the text but very difficult to keep track of. It would be immensely helpful if the authors added a Table with the following information for OS and OD for each patient at various time points post-surgery: BCVA improvement (use asterisks to note if LEA or ETDRS charts were used), ERG results, FST results. For any patient or any eye, if the measurement was not taken, please include a comment stating reason (e.g., non-compliance of patient, no follow-up visit etc.). This Table will nicely summarize the results and will be immensely helpful for the readers as well as for the field in general.

2.       Figure 2: Authors mention that the lesion shown at the injection site was first observed at 4 weeks post-surgery, however, images for only 23- and 32-months post-surgery are shown. If images are available, please include the 4-week images.

3.       Figure 5: The bar graphs only show the mean of responses from four patients. Please add data points for each patient for the bar graphs. This will be important to show the distribution across patients.

Minor comments:

1.       Line 503: “Since RPE65-linked IRD is a rod-dominated disease…..”. I understand what authors mean here. However, some readers may think that RPE65 mutation is in rods. It would be better to state something along the lines of ‘Since RPE65 mutations in RPE affect rod function…..’

2.       Line 154 states P2 is 4-year old male, and Line 157 states that P2 is 3 year old male

3.       Line 160 Best corrected visual acuity should be used as BCVA instead of VA. Sometimes VA is used sometimes BCVA is used. It would be good to keep the usage consistent throughout the manuscript.

4.       Table 3: Please list in the title or the legend that the data is combined data for all 8 eyes.

5.       Line 108: “…..and after 6 and 12 months.”  Would be better to state ‘…..and after 6 and 2 months of treatment’.

6.       Line 143: “…..regimen used in our children…..” Instead of ‘our children’, it would be better to state ‘in the children’ or ‘in the treated children’ or something similar.

Author Response

Gerhardt et al describe the results of Voretigene Neparvovec (RPE65 gene therapy) in four children afflicted with Leber congenital amaurosis (LCA). The follow-up results in these pre-school aged children showed improvement in vision and improved results across all retinal electrophysiology assessments. This is a well-written manuscript and provides important information about the responses and outcomes of RPE65 gene therapy in children. Following are the comments on the manuscript:

General response: we thank reviewer 1 very much for the detailed proofreading and valuable comments which helped us enormously to improve our manuscript.

Major comments:
1. One aspect that is challenging to following the manuscript is the data for individual patients and why some of the measurements are missing. Most of it is stated in the text but very difficult to keep track of. It would be immensely helpful if the authors added a Table with the following information for OS and OD for each patient at various time points post-surgery: BCVA improvement (use asterisks to note if LEA or ETDRS charts were used), ERG results, FST results. For any patient or any eye, if the measurement was not taken, please include a comment stating reason (e.g., non-compliance of patient, no follow-up visit etc.). This Table will nicely summarize the results and will be immensely helpful for the readers as well as for the field in general.

Response 1: We can understand this point of criticism very well and are thankful for the reviewer’s suggestion. In general, examinations are often very difficult in infants – in particular the younger the children are - due to their reduced attention span and compliance and the generally very young age of our patients, and this results in an incomplete data set. As per the request, we have created a table stating the examinations performed at different timepoints/available data and included it as a supplemental table (Suppl. Table 1). We very much hope that this meets the expectations of the reviewer. 

2. Figure 2: Authors mention that the lesion shown at the injection site was first observed at 4 weeks post-surgery, however, images for only 23- and 32-months post-surgery are shown. If images are available, please include the 4-week images.

Response 2: We thank the reviewer for careful reviewing the manuscript and picking up this issue. In fact, we had considered including the fundus image of the left eye taken 4 weeks post-surgery, but ultimately decided against it because of poor image quality. We have adjusted the figure at the request of the reviewer and hope that the retinal change is sufficiently visible despite the poor image quality, and that the figure is of satisfactory quality.

Figure 5: The bar graphs only show the mean of responses from four patients. Please add data points for each patient for the bar graphs. This will be important to show the distribution across patients.

Response 3: To show the individual responses of treated patients, we created a novel figure which is divided into two sub-figures (Suppl. Figure 3A and 3B). To avoid putting too much data into the original manuscript and to ensure the reading flow of the manuscript, we decided to put the new figure in the supplementary material and hope that this will also be appreciated by the reviewer. We can, of course, still change this and include it in the regular manuscript, if desired and are looking forward to receiving the reviewer’s feedback. 

Minor comments:
1. Line 503: “Since RPE65-linked IRD is a rod-dominated disease…..”. I understand what authors mean here. However, some readers may think that RPE65 mutation is in rods. It would be better to state something along the lines of ‘Since RPE65 mutations in RPE affect rod function…..’

Response: We have changed the phrase to “Since RPE65 mutations in RPE affect rod function even at earliest disease stages, …” 

2. Line 154 states P2 is 4-year old male, and Line 157 states that P2 is 3 year old male

Response: Thank you, we corrected this and apologize for the mistake (P1 is the 4 year old, and P2 the 3 year old). 

3. Line 160 Best corrected visual acuity should be used as BCVA instead of VA. Sometimes VA is used sometimes BCVA is used. It would be good to keep the usage consistent throughout the manuscript.

Response: Thank you, we corrected the manuscript accordingly (incl. updating the visual course figure). 

4. Table 3: Please list in the title or the legend that the data is combined data for all 8 eyes.

Response: Thank you very much for your comment. We corrected the legend according to your suggestion.

5. Line 108: “…..and after 6 and 12 months.” Would be better to state ‘…..and after 6 and 2 months of treatment’.

Response: Thank you very much for your comment. We corrected the passage according to your suggestion and added “of treatment”.

6. Line 143: “…..regimen used in our children…..” Instead of ‘our children’, it would be better to state ‘in the children’ or ‘in the treated children’ or something similar.

Response: We corrected the passage according to your suggestion to “in the treated children”

Reviewer 2 Report

Gerhardt et al. realized a very interesting article describing the “Gene therapy with Voretigene Neparvovec improves vision and partially restores electrophysiological function in pre-school children with Leber Congenital Amaurosis”. I consider the manuscript very interesting but, at the same time, I suggest several revisions needed to improve the reliability and the completeness of the paper: 

·      The “Introduction” and “Discussion” sections should be improved. I suggest comparing produced data with results obtained from other NGS studies involving related to oxidative stress and inflammation, which exert a pivotal role in Leber etiopathogenesis. The recent PMID: 32877751, PMID: 30523548 and PMID: 36290689 could represent a substrate able to enforce the role of considered cellular mechanisms.

·      Finally, manuscript requires English revisions and typos correction. 

Author Response

Gerhardt et al. realized a very interesting article describing the “Gene therapy with Voretigene Neparvovec improves vision and partially restores electrophysiological function in pre-school children with Leber Congenital Amaurosis”. I consider the manuscript very interesting but, at the same time, I suggest several revisions needed to improve the reliability and the completeness of the paper:
- The “Introduction” and “Discussion” sections should be improved. I suggest comparing produced data with results obtained from other NGS studies involving related to oxidative stress and inflammation, which exert a pivotal role in Leber etiopathogenesis. The recent PMID: 32877751, PMID: 30523548 and PMID: 36290689 could represent a substrate able to enforce the role of considered cellular mechanisms.
- Finally, manuscript requires English revisions and typos correction.

General Response:
Dear reviewer, 
we want to thank you very much for the positive feedback and the suggestions for improving our manuscript. 

Response to the first point:
Undoubtedly, oxidative stress and inflammation play an important role in the development and progression of retinal degeneration seen in inherited retinal diseases. These processes happen concomitantly to a pre-existing genetic defect that triggers the disease, and contribute to the further course of the disease. We noticed that you focused very much on basic pathophysiological mechanisms of retinal degeneration which we – as clinician-scientists ourselves – value high and appreciate. These are undoubtedly areas of research with highest relevance and many unanswered questions, and we regret that we have not been able to address these issues comprehensively in our work. In the following, we would like to explain why we have difficulties in adequately addressing your criticism. 

Since clinical retinal gene therapy is an exciting new field with limited real world data, our study’s focus was explicitly on the clinical description of the impact and efficacy of the recently approved subretinal gene therapy voretigene neparvovec in a largely unexplored patient cohort of pre-school children. There is plenty of existing basic research and published literature dealing with the pathomechanisms of retinal degeneration - particularly in the context of retinal dystrophies. Despite our existing knowledge, many underlying pathophysiological consequences and relationships remain a mystery and addressing these aspects would need extensive explanation. Therefore, a comprehensive discussion of pathophysiological correlations and explanatory attempts, which would do justice to all aspects of the underlying retinal degeneration in leber.

Reviewer 3 Report

The manuscript is considered as of a good quality, is well written and with a details description of the methods and obtained results. I have just a suggestion to improve the manuscript. Please add more details about the ERG recording performed including all the intensity of the light stimuli used. 

Author Response

Comments of Reviewer 3:
The manuscript is considered as of a good quality, is well written and with a details description of the methods and obtained results. I have just a suggestion to improve the manuscript. Please add more details about the ERG recording performed including all the intensity of the light stimuli used.

Response: We thank the reviewer very much for the positive feedback on our work. According to your suggestion, we updated the ERG chapter in the methods. We stated that 30Hz Flicker ERG was performed in a light-adapted state with an intensity of 2.5 cds/m2.

Further comment: due to the difficult examination conditions in children (e.g. reduced cooperation/compliance, faster fatigue, restlessness, refusal to undergo examinations), only the 30Hz flicker examination could be performed and thus testing of cone function was performed but no full ERG could be recorded.
